# Estimating BMI distributions by age and sex for local authorities in England: a small area estimation study

Ben Amies-Cull ©,[1] Jane Wolstenholme ©,[2] Linda Cobiac,[3] Peter Scarborough © [1]

¹Nuffield Department of Population Health, University of Oxford, Oxford, UK
²HERC, Nuffield Department of Population Health, University of Oxford, Oxford, UK
³School of Medicine, Griffith University, Nathan, Queensland, Australia

**Correspondence to**
Dr Ben Amies-Cull;
ben.amies@doctors.org.uk

## ABSTRACT

**Objectives** Rates of overweight and obesity vary across England, but local rates have not been estimated for over 10 years. We aimed to produce new small area estimates of body mass index (BMI) by age and sex for each lower tier and unitary local authority in England, to provide up-to-date and more detailed estimates for the use of policy-makers and academics working in non-communicable disease risk and health inequalities.

**Design** We used generalised linear modelling to estimate the relationship between BMI with social/demographic markers in a cross-sectional survey, then used this model to impute a BMI for each adult in locally-representative populations. These groups were then disaggregated by 5-year age group, sex and local authority group.

**Setting** The Health Survey for England 2018 (cross-sectional BMI data for England) and Census microdata 2011 (locally representative).

**Participants** A total of 6174 complete cases aged 16 and over were included.

**Outcome measures** Modelled group-level BMI as mean and SD of log-BMI. Extensive internal validation was performed, against the original data and external validation against the National Diet and Nutrition Survey and Active Lives Survey and previous small area estimates.

**Results** In 94% of age–sex are groups, mean BMI was in the overweight or obese ranges. Older and more deprived areas had the highest overweight and obesity rates, which were particularly in coastal areas, the West Midlands, Yorkshire and the Humber. Validation showed close concordance with previous estimates by local area and demographic groups.

**Conclusion** This work updated previous estimates of the distribution of BMI in England and contributes considerable additional detail to our understanding of the local epidemiology of overweight and obesity. Raised BMI now affects the vast majority of demographic groups by age, sex and area in England, regardless of geography or deprivation.

## INTRODUCTION

Good quality primary data on body mass index (BMI) do not exist for local areas of England. The sample size and sampling structure of national health surveys generally prevent health variables from being estimated at the local level by simple disaggregation, so

## STRENGTHS AND LIMITATIONS OF THIS STUDY

⇒ Social and demographic heterogeneity is incorporated into analysis by using the Census microdata 2011 and generalised linear model regression, allowing variation in exposure-and-effect relationships.
⇒ Extensive internal and external validation was performed in line with previous small area estimation studies.
⇒ Results required calibration, particularly affecting younger individuals in the microdata, to allow the distribution of body mass index (BMI) to closely reflect the base data.
⇒ It was not possible to estimate parameter uncertainty for these estimates.
⇒ Accounting for ethnicity-specific variation in BMI–risk relationships was not possible, for example, for the variation observed between ethnic groups in the risk of type-2 diabetes at a given BMI.

a wide variety of local area estimation techniques have been developed.[1–3] These are largely designed to predict the prevalence of a categorical variable rather than a continuous distribution that is more appropriate for many variables such as BMI. These estimates are useful for policymakers at local and national levels to understand where and for who the burden of raised BMI is highest and where to direct scarce resources. They also provide a valuable input to further research into BMI-related risk and related health inequalities.

The most recent set of estimates for BMI at the local authority level in England are by Moon *et al*[4] based on data from 1998 to 1999. Their paper used multilevel modelling to estimate the proportions of each local authority in the overweight and obese BMI categories. This found obesity rates were highest in parts of the West Midlands, Yorkshire and the Humber, and overweight rates highest in coastal areas, particularly in Devon and Cornwall.

Survey data exist for estimating risk factor prevalence in England from studies such as the Health Survey for England (HSE),[5] National Diet and Nutrition Survey (NDNS)[6] and Active Lives Survey (ALS),[7] but each of these has limitations that prevent them from giving precise estimates of risk factor prevalence for age and sex groups at the local authority level. Both the HSE and NDNS are underpowered for such estimates and they do not report each respondent's local authority. The ALS collects self-reported height and weight data from phone interview and reports data at the local authority level, but only reports BMI as BMI category (eg, overweight, obese). These surveys suffer with the common problems for small area estimation[8] that they do not specify the local geographic area of the respondent and their sample sizes are too small to deliver precise estimates of risk factors for subgroups. To overcome these problems with survey data, a wide variety of small area estimation methods have been developed to allow researchers to estimate the prevalence of risk factors for small geographical areas. These methods can be grouped into three broad approaches: indirect standardisation, model-based estimation and microsimulation approaches. The indirect standardisation approach involves calculating the proportion of the population with a feature of interest (eg, BMI in the overweight range) for different subgroups, then weights these proportions according to the subgroups' representation in each local area. Model-based estimation improves on this approach by using logistic regression (usually) to estimate the proportion of a feature in a local area based on individual-level and area-level variables, using aggregate data on the frequency of these variables.[1 2] This approach was improved on by using Census data to increase the number of variables available for use.[9] To better account for unmeasured variation between areas, for example, related to more granular sociodemographics, local context and culture,[10] Twigg et al[11] developed an approach using multilevel logistic regression (later used for the BMI estimates by[4 4] mentioned above). Finally, microsimulation approaches combine spatially aggregated survey data with geographically disaggregated microdata (that may need to be simulated)[8] but estimating credible intervals is challenging.[12 13]

We set out to estimate and validate the continuous distribution of BMI by age and sex in each local authority in England. We built on previous approaches to estimate continuous distributions, combining the strengths of model-based estimates and microsimulation.

## METHODS

Producing the modelled estimates of BMI distributions involved the following steps. First, we selected appropriate national-level survey data that included a BMI variable. Then we identified social and demographic variables that existed in both the health survey and the Census microdata 2011 (comprising every 20th entry from the UK Census 2011, available for all local authorities in England).[14] We then fit a model relating these variables with BMI on the health survey data and applied the model to each respondent in the Census microdata, to generate predictions of BMI at the individual level. These estimates were validated using established approaches.[12 15]

## Data

For the national-level survey data on BMI, the most recent wave of HSE (2018)[5] was chosen as the BMI dataset on the basis that the alternative of the NDNS 2014–2016 has a much smaller sample size than the HSE[6] and the ALS 2017 has only a self-reported categorical BMI variable.[7]

The local authority-level data were taken from the Census microdata 2011, which is a 5% sample of anonymised individual-level Census records, with the lowest geographical level at the lower tier or unitary local authority (UK local government, mostly 1 50 000–5 00 000 persons, of which there are 315 in England, plus the Isles of Scilly and City of London), or small groups of these termed 'local authority Census groups' where populations are small.

'Candidate variables' for the model were social or demographic markers that were available in the same or translatable forms in the HSE 2018 and the Census microdata. These were age, sex, ethnicity, self-rated health (from 1 (good) to 5 (poor)), number of cars in the household, socioeconomic status (using the National Statistics Socioeconomic Classification), NUTS-1 region (Nomenclature of Territorial Units for Statistics Level 1, namely the nine regions of England), tenure status (ie, living arrangements, such as renters) and dummy variables for being unemployed, being a student, having no formal qualifications, and having a degree. Index of Multiple Deprivation (IMD) is given in the HSE at the postcode level, but there is no IMD variable given in the Census microdata, so it had to be merged at the local authority level from an external dataset.[16] Online supplemental appendix table 1 provides a full description of these candidate variables and adaptations to make them equivalent. Participants in the HSE with missing data either for BMI or any of the candidate variables were dropped, detailed in online supplemental appendix tables 1 and 2.

## Generalised linear models

The estimation process involved building a generalised linear model (GLM) regression model in HSE 2018 with BMI as the dependent variable and the candidate variables as independent variables. This model was then used to predict a BMI for each individual in the UK Census 2011 microdata. To specify the GLM model, the data were first examined. The variance function (or family) was chosen on the basis of examination of the BMI variable, which identified a continuous skewed bell-shaped distribution, consistent with either a gamma or a gaussian (including log-normal) mean–variance relationship. Formal tests for family choice such as the modified Park test[17] could not be used as they rely on squared error and are not valid with clustered sampling. The Cullen-Frey

plot and QQ-plot in online supplemental appendix figures 1 and 2, respectively, indicate the BMI variable conformed more closely with a lognormal than gamma distribution. Therefore, the GLM was built specifying a gaussian family with log link, as has become an accepted practice[17 18] for log-normally distributed dependent variables. A frequency weight for the nurse visit was applied that was developed by the HSE team to account for non-response to the nurse visit, allowing representativeness to the general population. Cluster robust SEs were specified to account for clustered sampling in HSE.[19]

The model was built by regressing each variable against BMI for individuals aged 16 and over and including it in the model if Bayesian inference criterion (BIC) indicated an improved model (with an arbitrary cut-off of −2 to indicate superiority). BIC was used over the Akaike Information Criterion as BIC prefers more parsimonious models. Likelihood ratio tests were not used as they are invalid with robust standard errors.[16] Ordinal variables were assessed as continuous before assessing if the levels contributed beyond linearity. If the final model output Wald Test did not support a significant association (at p<=0.05), the variable's contribution was reassessed by BIC.

Outputs were used to impute BMI in the Census microdata for each respondent aged 16 and over. GLM log-link variables are given in the log scale and interpreted multiplicatively as:

$$E\left(Y\right) = \prod_{i=0}^{n} exp\left(\beta_i X_i\right)$$

Where E(Y) is the estimate of the independent variable (BMI), $X_i$ are variable values (where $X_0$=1 for the constant) and $\beta_i$ are coefficients. The process was as follows:

► Each individual's variable values were multiplied by their coefficients, then these were exponentiated, then multiplied together to give their final imputed BMI value.

► Each individual's coefficient values had random variation added to represent subgroup heterogeneity. This was done by adding random variation to these values on a normal distribution with SD equal to the SE of the coefficient from the GLM.

Calibration of the results was required to broaden the spread of data points to meet that in the natural population.

► For this, the SD of log-BMI was calculated for each age–sex group in the HSE data and these values were smoothed out by using a linear regression model to generate predicted values, using age and sex as independent variables.

► Uncalibrated individual estimates of log-BMI were disaggregated by age group and sex, then the mean of that group subtracted, results divided through by that group's SD to give a z-score.

► This was then multiplied up by the smoothed modelled SD from HSE and the group mean added back on, before exponentiating log-BMI back to BMI.

### Validation steps
Three validation steps were used, based on previous approaches.[12 15]
1. Estimates were assessed for heteroskedasticity by examining residual plots of log-BMI. Formal tests for heteroskedasticity (eg, Breush-Pagan test) are not valid with robust standard errors.
2. Internal validity was tested by comparing age-standardised distributions of estimated and measured BMI (from HSE 2018) by sex.
3. External validity was tested in three ways.
► Comparing age-standardised distributions of estimated and measured BMI from NDNS 2014–16 by sex.
► For each local authority, the estimated obesity rate was plotted against the obesity rate according to self-reported BMI from the ALS 2017.
► Results were compared with[4] in terms of the areas in the top decile for obesity rates and for overweight rates.

### Public and patient involvement statement
Patients or the public were not involved in the design, or conduct, or reporting, or dissemination plans of our research.

### RESULTS
BMI was estimated for adult males and females across 16 age groups in 249 local authority census areas of England, making 7968 subgroups. In 94% of these subgroups the mean BMI was in the overweight range (97% for males, 90% for females). In no area was mean BMI in the normal range between the ages of 50 and 85 years for either sex. Mean BMI, mean log-BMI and SD of log-BMI by age group, sex and local area are available from Oxford Research Archive. Online supplemental appendix table 4 shows that people with raised BMI in HSE 2018 tended to be older and more deprived and had worse health. There were 6174 complete cases in HSE and online supplemental appendix table 2 shows details of missing data, identifying 18% missing for the BMI variable and online supplemental table 3 shows that older people are overrepresented in these missing data.

### Model results
Table 1 shows the coefficients for each of the variables fitted in the final model, with their 95% CIs.

After standardising for age, mean BMI was 27.7 kg/m$^2$ for females and 27.9 kg/m$^2$ for males, though the distribution was broader for females, with higher proportions in low and high categories. Figure 1 shows the orders of magnitude of difference between local authorities across different quintiles of deprivation with the England average for reference. Even for the lowest deprivation area, almost all age–sex groups have mean BMI in the overweight range.

**Table 1** Model variables, coefficients and coefficient 95% CIs given to three significant figures

| Variable | Coefficient (log scale) | 95% CIs Lower | Upper |
|---|---|---|---|
| Age (years) | 0.0114 | 0.00932 | 0.0136 |
| Age (years) squared | $-9.81\times10^{-05}$ | −0.000119 | $-7.70\times10^{-5}$ |
| Female (dummy) | 0.108 | 0.0288 | 0.188 |
| Female×age | −0.00398 | −0.00709 | −0.000874 |
| Female×age squared | $3.22\times10^{-5}$ | $3.66\times10^{-6}$ | $6.08\times10^{-5}$ |
| Ethnicity White/mixed (base) | | | |
| Black | 0.0483 | 0.0147 | 0.0819 |
| Asian | −0.0219 | −0.0435 | −0.000323 |
| Other | −0.0423 | −0.0901 | 0.00538 |
| Self-employed (dummy) | −0.0159 | −0.0337 | 0.00200 |
| Retired (dummy) | 0.00450 | −0.0154 | 0.0244 |
| Long-term sick (dummy) | −0.0221 | −0.0653 | 0.0210 |
| No qualifications (dummy) | −0.00621 | −0.0209 | 0.00844 |
| Degree holder (dummy) | −0.0286 | −0.0407 | −0.0164 |
| IMD Quintile 1 (base, lowest deprivation) | | | |
| Quintile 2 | 0.00732 | −0.00890 | 0.0235 |
| Quintile 3 | 0.0130 | −0.00446 | 0.0305 |
| Quintile 4 | 0.0176 | 0.000301 | 0.0349 |
| Quintile 5 | 0.0315 | 0.0116 | 0.0514 |
| Health status Very good (base) | | | |
| Good | 0.0574 | 0.0467 | 0.0682 |
| Fair | 0.108 | 0.09207 | 0.124 |
| Bad | 0.109 | 0.0794 | 0.138 |
| Very bad | 0.142 | 0.0838 | 0.200 |
| Tenure status Own outright (base) | | | |
| On a mortgage | −0.00130 | −0.0178 | 0.0152 |
| Shared ownership | 0.0275 | −0.0525 | 0.107 |
| Rental | 0.0192 | 0.00304 | 0.0354 |
| Living rent-free | 0.0684 | −0.000499 | 0.137 |
| Region North East (base) | | | |
| North West | −0.00122 | −0.0249 | 0.0225 |
| Yorks and Humber | 0.0209 | −0.00582 | 0.0477 |
| East Midlands | 0.00591 | −0.0194 | 0.0312 |
| West Midlands | 0.0202 | −0.00633 | 0.0467 |
| East of England | 0.0122 | −0.0140 | 0.0384 |
| London | −0.0289 | −0.0540 | −0.00373 |
| South East | 0.0162 | −0.00975 | 0.0423 |
| South West | −0.00181 | −0.0259 | 0.0222 |

Continued

**Table 1** Continued

| Variable | Coefficient (log scale) | 95% CIs Lower | Upper |
|---|---|---|---|
| SES AB—managerial and professional (base) | | | |
| C1—intermediate occupations | 0.00200 | −0.0177 | 0.0217 |
| C2—Small employers and own account workers | 0.0141 | −0.0113 | 0.0395 |
| DE—lower supervisory, technical, and semi-routine occupations | −0.0116 | −0.0280 | 0.00480 |
| Other/no response | −0.0123 | −0.0366 | 0.0120 |
| Constant | 2.97 | 2.91 | 3.03 |

IMD, Index of Multiple Deprivation; SES, socioeconomic status.

## Validation

### Heteroskedasticity

Residual plots (online supplemental appendix figure 3) demonstrated homoskedasticity. Trend lines show no overall bias across measured log-BMI and residuals' are shown to be normally distributed in online supplemental appendix figures 4A and 4B.

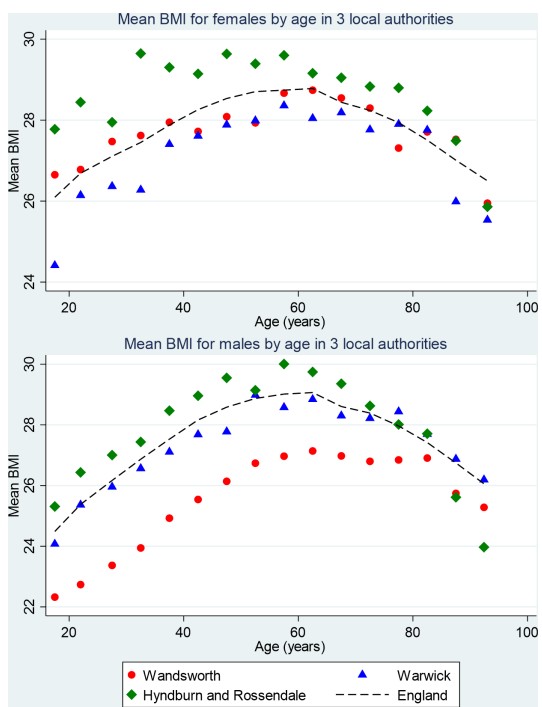

**Figure 1** Distribution of BMI across the age range for three local authorities for females (A, above) and males (B, below): Manchester (IMD quintile 5—highest deprivation), Oxford (IMD quintile 3) and South Gloucestershire (IMD quintile 1). BMI, body mass index; IMD, Index of Multiple Deprivation.

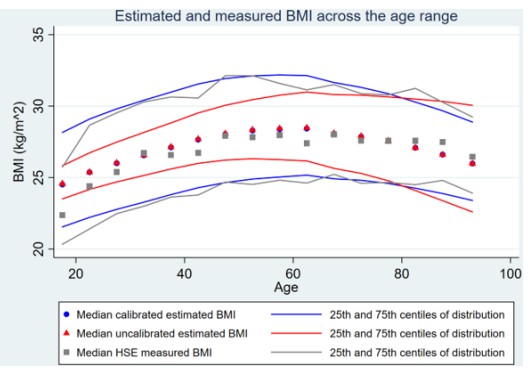

**Figure 2** Scatter plot of mean measured BMI, uncalibrated estimate and calibrated estimate, with their associated 25th–75th centile range across the age range (for both sexes combined). BMI, body mass index; HSE, Health Survey for England.

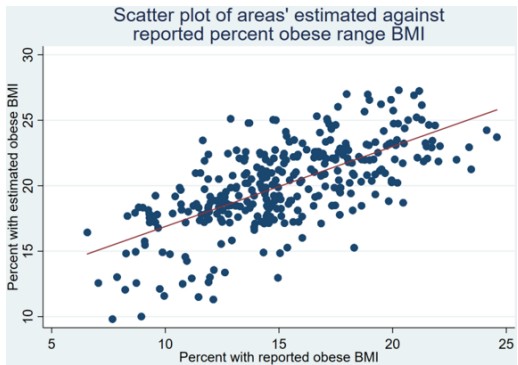

**Figure 4** Scatter plot of local areas' age-standardised percent of population with estimated BMI in the obese range against self-reported BMI in obese range in ALS 2017, with best fit line. ALS, Active Lives Survey; BMI, body mass index.

### Internal validity

The age-standardised England-level distributions of estimated BMI for females and males are compared with the equivalent distributions of measured BMI from the original HSE 2018 data, shown in online supplemental

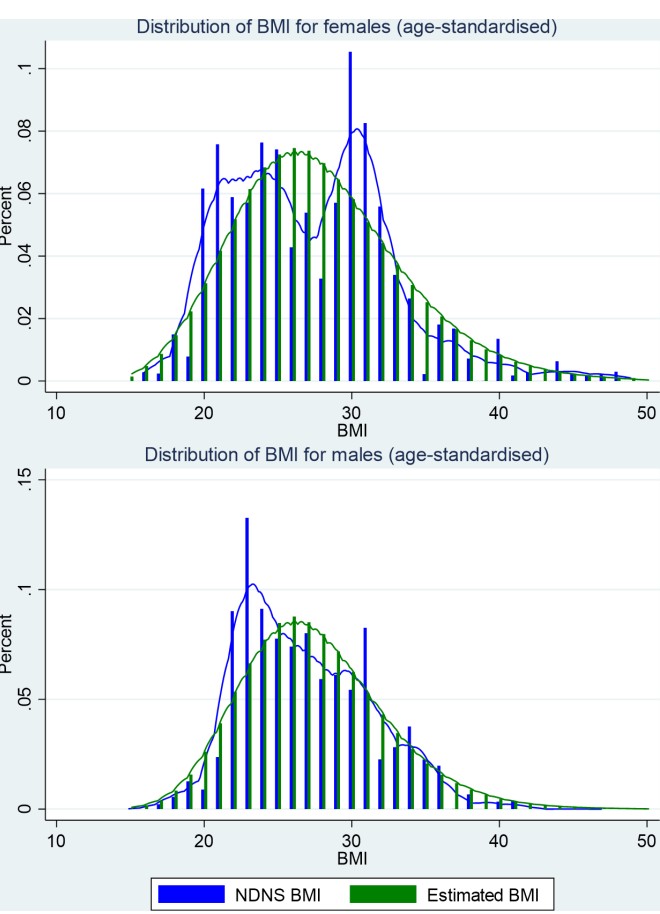

**Figure 3** Histogram of age-standardised BMI estimates for England for females (A, above) and males (B, below), including the equivalent histogram for measured BMI from NDNS 2014–2016 for comparison. BMI, body mass index; NDNS, National Diet and Nutrition Survey.

appendix figures 5A and 5B, showing close concordance with the base data.

Figure 2 compares the 25th, 50th and 75th centiles of the calibrated and uncalibrated estimates with the HSE data across the age range across England, standardising the populations to represent females and males equally. These show that calibration mainly affected younger age groups and produced distributions closely in alignment with those in the HSE.

### External validity

Estimated BMI distributions for females and males were compared with the equivalent distributions from the independently collected NDNS 2014–2016 data in figure 3A, B (standardised to represent each age group equally). Figure 4 plots the reported proportion of people with BMI in the obese range for each local authority in the ALS against the same proportions on the new estimates. This shows a close relationship, though the proportion in the obese range is approximately 5 percentage points higher in the estimates than ALS across the distribution. Online supplemental appendix figures 6A–6D show graphs for the other BMI categories.

Comparison of model results with those by Moon *et al*[44] shown in figure 5. These maps would not be expected to be the same as the data they are estimated on are 20 years apart. However, where specific areas are not the same, adjacent and similar types of areas are highlighted, namely mainly Yorkshire and the Humber, the West Midlands, coastal and isolated rural areas. Coefficients used in their model were similar to this paper: individual-level age, sex, black ethnicity, Asian ethnicity and area-level percentages of low social grade, high social grade, households with dependent children, in local authority housing and adults married. Likewise, their greatest coefficients were with age and sex, but black and Asian ethnicities were more important in their paper while health status is more influential in these new estimates.

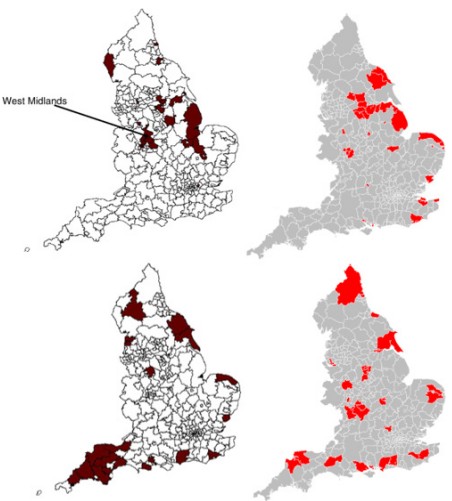

West Midlands

**Figure 5** Comparison of results from Moon *et al*[4] (left) with new estimates (right), showing the top decile of areas with the highest rates of obesity (above) and overweight (below). Left images reprinted from Moon *et al*[25]

## DISCUSSION

This paper introduces a method that combined the individual-level associations of BMI with social and demographic features to estimate the BMI of each individual in the Census microdata 2011, allowing a mean BMI and associated distribution to be estimated for each 5-year age and sex group for each local authority. It builds on the established methods estimating local BMI[1 4 9 20] while adding further detail by estimating continuous distributions of BMI. These will provide policy-makers and academics much greater detail on BMI-related risk, allowing policy impact on health inequalities to be better understood and enabling resources to be better targeted.

The distributions of age-level and sex-level estimates were calibrated to replicate the national distribution of BMI. Figure 2 shows that this process produced highly consistent distributions with the underlying data and calibration affected younger ages groups more. Females had higher proportions with low and high BMI. BMIs were higher in older and more deprived areas.

Examination of residuals identified homoskedasticity (online supplemental appendix figure 2), in the context of the GLM implying that the link function was appropriately specified and that the model was well fitted. Internal validation against HSE showed good concordance. External validation at the local authority level against the ALS in figure 4 showed a close linear relationship between the proportion of the population in the obese range, though with a distribution approximately 5 percentage points higher in the estimates. This could be due to the reporting bias of the self-reported height and weight items in the ALS or response bias of people with lower BMI into the survey. The other internal and external validation exercises

indicate that this is unlikely to be a consequence of bias in the estimates.

Comparison of the new estimates against those from[44] show close alignment. Many of the same local areas are identified, such as Birmingham, Humberside, Lincolnshire and the south coast. In others, if the same area was not identified then neighbouring areas usually were, such as around Norwich.

### Strengths and limitations

This novel approach of estimating BMI at the individual level has allowed estimates to be drawn with a greater degree of detail than previously, namely estimating the whole BMI distribution and drawing estimates for many subgroups. The HSE and Census microdata are both highly granular and representative of the population and data quality of the HSE is generally good, being a large and representative sample. One relevant limitation to the HSE 2018 is missing data, namely for the BMI variable, especially from the late 70s (online supplemental appendix table 3). One potential reason that older people are underrepresented may be poor health. While it is not possible to account for systematic differences in non-responders, health is captured in the model, while the Census should provide a representative estimate of health in each age group, helping to avoid bias from these missing data.

The range of variables that can be aligned between datasets is also a limitation, in common with previous local area estimation methods.[11] First, the levels of geography do not align apart from at the NUTS-1 level. This means that multilevel modelling previously used[4 11 21 22] is not feasible, as the NUTS-1 areas are far too large and internally heterogeneous to pick up local authority-sized variation. Therefore, this method assumes that the national-level associations between the predictors and BMI do not systematically vary between local areas. Second, deprivation is measured at the postcode level in the HSE 2018 but is merged at the local authority level into the Census data. This means that deprivation is not heterogeneous within local authorities in the Census as it is in the real world, so resulting in slightly narrower modelled BMI. A calibration step was used to allow estimated distributions to closely reflect the real world, relying on an assumption that the order of age-sex-area-level estimates produced by the model was correct.

The Census data are, of writing, 10 years old. Ageing populations are not directly important as age is accounted for, but local changes (such as increasing education levels) may lead to bias in estimates. As the model was built using recent HSE and IMD data, change in association between variables and BMI over time should not be a source of bias. The final limitation is that estimating the parameter uncertainty to a variable's estimated mean in small area estimation methods is difficult[8 12] and not attempted here.

The use of BMI as a predictor of health outcomes has been critiqued on the basis of heterogeneity in the population, particularly around ethnic differences. Different BMI cut-offs for disease risk have been described, for example, identifying that for the same risk of developing type-2 diabetes as white people with a BMI of 30 kg/m$^2$, South Asians only needed a BMI of 23.9 kg/m$^2$ and black people a BMI of 28.1 kg/m$^2$.[23] While it remains unclear why this pattern has arisen,[23] this correction could make a meaningful difference to disease risk in some areas with greater proportions of non-white people. Therefore, these results may need to be interpreted with such variation in mind and this topic remains an important focus for future research.

## CONCLUSION

This study used HSE 2018 to estimate the BMI of respondents in the Census microdata 2011, providing the most recent set of disaggregated small area estimates for BMI at the local authority level in England since 2007.[4] Granular social and demographic information provided by the Census microdata 2011 captured local variation. Results showed that over 50% of people in almost every demographic group and area had BMI above-the healthy range.

Validation showed there was a low risk of heteroskedasticity and there were approximately normally distributed residuals, providing confidence that models were well fitted. Comparison of estimates against external datasets at the national and regional levels identified close concordance with previous findings, but comparison at the local authority level with self-reported BMI implied that many areas have higher rates of raised BMI than previously[24] estimated.

These new estimates go further than previous studies by two dimensions—by estimating BMI by age–sex groups and by modelling the distribution of BMI rather than percentages in BMI categories. As such, this study has produced the most demographically granular estimates for BMI in local areas of England to date.

**Contributors** BA-C conceived of the study, contributed to study design, undertook the main analysis andcompleted the first draft of the manuscript. JW conceived of the study, contributed to design, supervised the analysis and commented on drafts of the manuscript. LC conceived of the study, contributed to design, supervised the analysis and commented on drafts of the manuscript. PS conceived of the study, contributed to design, supervised the analysis and commented on drafts of the manuscript. BA-C acts as guarantor

**Funding** This work was funded by the Medical Research Council grant number MR/N013468/1.

**Disclaimer** Funders had no involvement in the study design; in the collection, analysis and interpretation of the data; in the writing of the report; or in the decision to submit the paper for publication.

**Map disclaimer** The depiction of boundaries on this map does not imply the expression of any opinion whatsoever on the part of BMJ (or any member of its group) concerning the legal status of any country, territory, jurisdiction or area or of its authorities. This map is provided without any warranty of any kind, either express or implied.

**Competing interests** BA-C is supported by the MRC, JW by the NIHR and CRUK, and PS by the BHF and NIHR Oxford Biomedical Research Centre.

**Patient and public involvement** Patients and/or the public were not involved in the design, or conduct, or reporting, or dissemination plans of this research.

**Patient consent for publication** Not applicable.

**Ethics approval** The terms of the Central University Research Ethics Committees were reviewed. The study did not require ethics approval as only pre-collected data on non-identifiable individuals was used.

**Provenance and peer review** Not commissioned; externally peer reviewed.

**Data availability statement** Data are available in a public, open access repository. Data are available at https://doi.org/10.5287/bodleian:JbamnGwNG.

**ORCID iDs**
Ben Amies-Cull http://orcid.org/0000-0002-7079-1288
Jane Wolstenholme http://orcid.org/0000-0001-7493-1850
Peter Scarborough http://orcid.org/0000-0002-2378-2944

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
