## [Reviewer comments · BMJ Open]

ARTICLE DETAILS

TITLE (PROVISIONAL)	Estimating BMI distributions by age and sex for local authorities in England: a small area estimation study
AUTHORS	Amies-Cull, Ben; Wolstenholme, Jane; Cobiac, Linda; Scarborough, Peter

VERSION 1 – REVIEW

REVIEWER	Hon Yiu So Oakland University, Mathematics and Statistics
REVIEW RETURNED	21-Feb-2022

GENERAL COMMENTS	The manuscript is generally well written. Major comments 1. As the title suggest, the author estimates the BMI distribution by age and sex for local authorities. Where can the ready find the estimated BMI distribution?2. Figure 2 shows the centiles of BMI by age, but it does not separate different sexes. How do estimate the centiles and the medians?3. How can you handle the survey weights?4. The central result states, "In 94% of these subgroups, the mean BMI was in the overweight range (97% for males, 90% for females). In no area was mean BMI in the normal range between the ages of 50 and 85 years for either sex" What are the subgroups? How many subgroups are there?5. In the conclusion, the authors say, "Results showed that over 50% of people in almost every demographic group and area had above-normal BMI." Which tables or figures support such results?6. In Figure 5, the result from this manuscript is quite different from Moon et al. 2007. Is there any reason for that?7. On page 33, the empirical BMI for females from NDNS indicates the BMI should be double-peaked. Can the model be modified to accommodate the mixture distribution? Minor comments: Page 26-27, and 33: the "histograms" are bar charts. Page 32: instead of plotting mean BMI, consider boxplots that contain more information. Page 32-36: texts like "154x169mm (85 x 85 DPI)" should be removed.
--

REVIEWER	Carmen Tekwe Indiana University Bloomington
REVIEW RETURNED	26-Mar-2022

GENERAL COMMENTS	Evaluation Summary: The manuscript by Amies-Cull et al. provides updated estimates of small area estimates of BMI by age and sex for lower tier and unitary local authority in England. The authors used GLM to examine the association between BMI with social and demographic markers in the first stage, and used the model to impute BMI for each adult in locally- representative populations at the second state. Some recommendations to the authors: Overall, the manuscript addresses an important topic in obesity, small area estimation, demography. The manuscript was also very well written and easy to follow. I have the following additional comments for the authors:  1. While BMI is often used as an indicator for adiposity, its adequacy for this purpose has been questioned especially for racial ethnic minorities. Is it possible to use an adjusted BMI? For example, see Caleyachetti et al. 2021 (Lancet). 2. Can the authors provide the estimation steps discussed (page 7, lines 34-44) in an itemized format so the readers can more easily follow the steps taken in their approach?
--

VERSION 1 – AUTHOR RESPONSE

Reviewer: 1

Dr. Hon Yiu So, Oakland University

Comments to the Author:

The manuscript is generally well written.

Thank you for your comments.

Major comments

1. As the title suggest, the author estimates the BMI distribution by age and sex for local authorities. Where can the ready find the estimated BMI distribution?

Thank you for your interest in the data, we have been preparing a repository item at the University's archive that is now ready at <https://doi.org/10.5287/bodleian:JbamnGwNG> and this link has been included in the data sharing statement of the manuscript.

2. Figure 2 shows the centiles of BMI by age, but it does not separate different sexes. How do estimate the centiles and the medians?

Thanks for highlighting this. The figure was prepared as a merged population to simplify the figure and limit the number of figures in the manuscript. The population has been standardised to represent females and males equally. We have amended the text to highlight this.

3. How can you handle the survey weights?

The nurse visit frequency weight was applied for GLM regression variable selection and this detail has now been added to the methods section.

4. The central result states, "In 94% of these subgroups, the mean BMI was in the overweight range (97% for males, 90% for females). In no area was mean BMI in the normal range between the ages of 50 and 85 years for either sex" What are the subgroups? How many subgroups are there?

The subgroups were the 16 age groups, 2 sexes and 249 local authority census areas, making 7968 subgroups. We hope the wording in the manuscript is clearer on this now.

5. In the conclusion, the authors say, "Results showed that over 50% of people in almost every demographic group and area had above-normal BMI." Which tables or figures support such results? Due to the volume and complexity of the data, we felt it would be too burdensome to include a table laying out this level of detail in the main paper. However, now you have identified this issue we have added a table to the Oxford Research Archive submission at <https://doi.org/10.5287/bodleian:JbamnGwNG> to fully demonstrate mean BMI by age, sex and local area (as well as by log-BMI).

6. In Figure 5, the result from this manuscript is quite different from Moon et al. 2007. Is there any reason for that?

Thank you for pointing this out. There are two sets of issues here. First and foremost, the method is different and the data that estimates are based on are separated by 20 years, so time moving on would predictably change the specific areas highlighted. Secondly, you rightly identify that we have

not accounted for the nuance that will be more implicit to British readers. Specifically, although the areas highlighted are not always the same, the types of area are very similar. We hope the added explanation in the manuscript helps account for this.

7. On page 33, the empirical BMI for females from NDNS indicates the BMI should be double-peaked. Can the model be modified to accommodate the mixture distribution?

The NDNS sample is small (1,500 adults) and neither theory nor other surveys would support a bimodal distribution in the natural population, for example, the Health Survey for England (used to build the model) has a sample size of 5,200 and demonstrates a more classical skewed bell-shaped curve for both sexes. Therefore we have approached model building assuming the true underlying distribution to be unimodal. Plotting the estimates against NDNS is to compare for similarity, granting the differences in the two surveys.

Minor comments:

Page 26-27, and 33: the “histograms” are bar charts.

- We have plotted the distributions against percent, so although the figures could be described with either term, we prefer histogram.

Page 32: instead of plotting mean BMI, consider boxplots that contain more information.

- To keep the figures simple, we opted for means, choosing to represent spread in figures 2, 3 and 4.

Page 32-36: texts like “154x169mm (85 x 85 DPI)” should be removed.

- Thanks for the attention to detail here. We have checked and these were not part of the original image files, so must have been added by the journal portal.

Reviewer: 2

Dr. Carmen Tekwe, Indiana University Bloomington

Comments to the Author:

Evaluation Summary: The manuscript by Amies-Cull et al. provides updated estimates of small area estimates of BMI by age and sex for lower tier and unitary local authority in England. The authors used GLM to examine the association between BMI with social and demographic markers in the first stage, and used the model to impute BMI for each adult in locally representative populations at the second state.

Some recommendations to the authors: Overall, the manuscript addresses an important topic in obesity, small area estimation, demography.

The manuscript was also very well written and easy to follow. I have the following additional comments for the authors:

Thank you for your comments and we hope these responses are satisfactory.

1. While BMI is often used as an indicator for adiposity, its adequacy for this purpose has been questioned especially for racial ethnic minorities. Is it possible to use an adjusted BMI? For example, see Caleyachetti et al. 2021 (Lancet).

Thanks for this really interesting observation and commentary. Applying an adjustment would not be possible at present as different diseases would need different adjustments (many of which may not have been described yet) so a balanced adjustment would have to be applied across adiposity-related diseases. However, such an adjustment could make a meaningful difference in some areas, so we have added a paragraph to the discussion to address these issues and included this in the bullet pointed limitations at the top of the manuscript. This is an important and novel focus for future work in small area estimation.

2. Can the authors provide the estimation steps discussed (page 7, lines 34-44) in an itemized format so the readers can more easily follow the steps taken in their approach?

Sorry if this section wasn't as clear as it needed to be and we hope the edits are an improvement.

VERSION 2 – REVIEW

REVIEWER	Hon Yiu So Oakland University, Mathematics and Statistics
REVIEW RETURNED	12-May-2022
GENERAL COMMENTS	I am happy that all the issues are well addressed.